# *¿Qué Pasa Con Papá?* Exploring Paternal Responsibilities and Physical Activity in Mexican-Heritage Families

**DOI:** 10.3390/ijerph18168618

**Published:** 2021-08-15

**Authors:** Megan E. McClendon, M. Renée Umstattd Meyer, Tyler Prochnow, Kelly R. Ylitalo, Andrew R. Meyer, Christina N. Bridges Hamilton, Joseph R. Sharkey

**Affiliations:** 1Texas A&M AgriLife Extension Service, Texas A&M University, College Station, TX 77843, USA; megan.mcclendon@ag.tamu.edu; 2Department of Public Health, Robbins College of Health and Human Sciences, Baylor University, Waco, TX 76706, USA; tprochnow@tamu.edu (T.P.); Kelly_Ylitalo@baylor.edu (K.R.Y.); cbridgeshamilton@brockport.edu (C.N.B.H.); 3Department of Health & Kinesiology, Texas A&M University, College Station, TX 77843, USA; 4Department of Health, Human Performance & Recreation, Robbins College of Health and Human Sciences, Baylor University, Waco, TX 76706, USA; Andrew_Meyer@baylor.edu; 5Department of Public Health & Health Education, SUNY Brockport, Brockport, NY 14420, USA; 6Department of Health Promotion & Community Health Sciences, School of Public Health, Texas A&M University, College Station, TX 77843, USA; jrsharkey@tamu.edu

**Keywords:** family centered, Latino/a, Mexican-heritage, co-participation, physical activity

## Abstract

Mexican-heritage children often achieve less physical activity (PA) than their counterparts and are at greater risk for associated comorbidities. Child PA is greatly influenced by their parents, yet researchers have rarely involved fathers in community health promotion. The purpose of this study is to examine Mexican-heritage fathers’ perceptions of responsibilities and self-reported activities. *Promotoras* recruited fathers (*n* = 300) from colonies on the Texas–Mexico border and administered Spanish-language surveys including paternal responsibilities, father PA, and PA co-participation. Two researchers coded responses. Open-ended items were coded and cross-tabulations between responsibilities and activities with children were examined. Fathers reported feeling monetary responsibilities most often. Fathers reported engaging in more activities with their sons than daughters; however, fathers engaged in very few activities specifically with their children. Feeling responsible for family expenses was associated with paternal PA co-participation with family and children. This study adds clarity to the role of Mexican-heritage fathers in child PA. Findings highlight potential areas for intervention including supporting fathers to take an active role in their children’s PA.

## 1. Introduction

Physical activity (PA) is associated with improved physical, mental, and overall health and reduced risk for chronic diseases [1,2,3,4]. More specifically, physical inactivity is associated with childhood diabetes [5], cancer [4], cardiovascular disease and all-cause mortality [3]. Children ages 6 to 19 are recommended to attain 60 min per day of moderate to vigorous PA including muscle strengthening activities on at least three days per week [1,6]. Yet, fewer children achieve PA recommendations as compared to past years [7]. Further, rural communities experience disparities in PA participation often associated with unique barriers, including limited resources and amenities, limited transportation, and long distances to resources which disproportionally affect rural communities [8,9,10,11].

Unfortunately, disparities based on race and ethnicity have also been reported in child PA as well as related health outcomes. Specifically, Latino/a populations experience disproportionate rates of obesity [12], poor dietary habits [13], diabetes [14], physical inactivity [15], and screen time [15] when compared to other racial and ethnic groups. These disparities often times are related to one another and function as comorbidities. Given noted disparities, researchers must pay specific attention to understand cultural nuances in these populations. Further, individuals of Mexican-heritage integrate Latino/a and American culture which can negatively affect interaction with their host culture due to reduced acculturation (adoption of cultural practices and ideas) [16]. In this way, adoption and adaptation of negative health behaviors linked to either the cultural practices of their traditional culture or their host culture can create obesogenic environments or unhealthy behaviors [16]. This potential negative interaction creates the need to prioritize Mexican-heritage families and given supporting evidence, involve parents as agents of change in their child’s PA [17].

Past intervention attempts to improve child PA have targeted different settings; mostly school [18] and home-based [19]. In a national sample, parents exhibited influential factors regarding child PA outcomes (e.g., organized and leisure time PA) [17] leading to avenues of potential change. Parental and family influence has been established as a vector of change for child behaviors [17]. However, the majority of child PA research has involved maternal participation with disproportionate lower involvement and recruitment of fathers [20]. A recent review was conducted to highlight the lack of paternal involvement in child PA research [20]. In this review, researchers revealed that only 10 studies included the voice of the father with all 10 consisting of cross sectional self-reported parenting practices and activity, and only four being conducted in the United States. Of these four, half were primarily focused on maternal participation [21,22], two had less than 20% Latino/a participation [22,23], one did not report parent sex [21], and one did not report race or ethnicity [24]. This dearth of knowledge underlines the need to understand the role of Latino and Mexican-heritage fathers.

Specifically, a focus group of 26 Latino fathers identifying as Mexican-heritage reported they set expectations for their child’s PA, provided logistic support for PA engagement (monetary and transport), and used role modeling as well as monitoring as ways to increase child PA [25]. While this study is one of the first to examine the perspectives of the father within a Latino ethnic population, it does not specifically speak to the Mexican-heritage population who could potentially adhere to different cultural practices [16]. These practices differ widely and are important to consider. For example, in a study investigating Mexican-heritage family engagement, mothers gave fathers low family engagement scores [26], with researchers citing separate gender roles as a contributing factor [27].

More research is needed to understand Mexican-heritage fathers, their feelings of responsibility in their families, and how it is related to child PA. Therefore, the purpose of this article is to examine Mexican-heritage father’s perceptions of their responsibilities within the home and activities with family members including co-participation in PA with their child (ren). As a descriptive and exploratory study, this article will provide vital information from the father’s point of view which is currently lacking.

## 2. Materials and Methods

### 2.1. Procedures

*Promotoras de salud* (or simply *promotoras*) recruited Mexican-heritage fathers (*n* = 300) to complete surveys. *Promotoras* are female community health workers who are members of the community, trained to conduct research, and advocate for their communities [28]. While male *promotores* have also been used in health programing, this project benefited from the female *promotoras* in this community employed on the project [28]. For recruitment, *promotoras* utilized a list of participants recruited for a prior project. To participate in this recruitment, fathers needed to have children between 8–10 years of age living at home, father who lived with his spouse/partner, have reliable transportation, preferred to read and write in Spanish, and were of Mexican-heritage (themselves, children, spouse/partner, mother, father, or in-laws were born in Mexico). Surveys were collected in the summer and fall of 2017 at a time and location convenient to the father. IRB approval and consent were obtained before survey collection began (IRB2014-0825D).

### 2.2. Sample

All participating fathers were of Mexican heritage, lived with their spouses, preferred Spanish in written documents and conversation, and had at least one child 8–10 years of age. Fathers resided in Hidalgo County, Texas which has over 860,000 residents of which 91.8 percent identify as Latino/a. Census data show over half of residents were employed (58.8 percent) which is lower than state average (64.3 percent). Additionally, median household income in Hidalgo County is substantially lower than state average ($37,106 versus $59,206 respectively), and Supplemental Nutrition Assistance Program (SNAP) participation was much higher than the state average (30.5 percent versus 12.0 percent respectively) [29].

### 2.3. Measures

Two open-ended prompts were used to explore the research questions of this study. First, fathers were asked “As a dad, what are your responsibilities at home?” Fathers could report as many responsibilities as they wished but were only required to report one. Inductive qualitative coding processes were used to create a framework of responsibilities. Researchers identified two parent codes (monetary and family) and 13 child codes. Table 1 provides the codes used for responsibilities.

Fathers were also asked, “What things do you actually do in your spare time?” This second question included follow-up prompts such as where, when, with whom, and how often. What and with whom were analyzed to determine if fathers spent time with children. These prompts were also used to determine if that time was spent doing PA or sedentary activities with their children. Survey construction allowed fathers to report up to four activities they do in their spare time and participants in those activities. Within activities reported, researchers identified two parent codes (physically active and sedentary activities) and sixteen child codes. Further details on codes generated are available in Table 2.

### 2.4. Statistical Analysis

Researchers utilized a grounded theory approach to identify themes and concepts related to the responsibilities and activities of Mexican-heritage fathers, an under-researched population [30]. Grounded theory was used in the absence of previous studies to develop a foundation for future theory to be built. Constructivist grounded theory is a secondary variant of grounded theory that allow researchers to interpret responses based on local culture [31]. Researchers conducted content analysis [32] before moving into inductive analysis that led to the identification of themes [33]. A primary researcher completed an initial content analysis to create a working coding scheme that included parent and child codes [31]. A secondary researcher acted as an external auditor to strengthen the validity of the codebook [31]. Reliability was determined by two researchers each coding segments of 10 percent of the surveys (*n* = 30) [31]. After reaching consensus both researchers coded an additional 10 percent of the remaining surveys that were selected randomly (*n* = 30). Intercoder agreement was set at 80 percent agreement on 95 percent of the codes [31] and was examined using Excel. After establishing inter-coder agreement, frequencies and cross-tabulations were calculated using SPSS v. 24 (IBM Corp., Armonk, NY, USA) [34].

## 3. Results

### 3.1. Responsibilities

Most fathers reported a second responsibility (*n* = 222; 74%), with fewer reporting three (*n* = 90; 30%) or four (*n* = 15; 5%). Of the 627 total reported responsibilities, 55.8% were monetary responsibilities with the most frequently reported responsibility being family expenses (*n* = 164/627; 26.2%). Reported family expenses included things like supplying “food and clothing for the children,” and working to “maintain the family.” The second most frequently reported responsibility was childcare (*n* = 88/627; 14.0%). Most paternal reported responsibilities contained similar phrasing across fathers, with “work, bring the food, advise the children, pay bills” coded as family guidance, monetary bills, monetary family expenses, and monetary general, respectively. “Take care of the children well and provide the food and clean the yard” was coded as family childcare and family household tasks respectively; “maintain the home and my children healthy” was coded as monetary home and family health, respectively. Table 3 provides complete reporting of frequencies of paternal responsibilities.

### 3.2. Paternal Reported Activities

Fathers reported most activities as either outside chores (*n* = 359/866; 41.4%) or inside chores (*n* = 141/866; 16.2%) chores. Outside chores included “mow the yard,” “plant trees,” and “maintenance of cars.” Fathers most frequently reported doing activities with their family (*n* = 344/866; 39.7%) and alone (*n* = 274/866; 31.6%). After stratifying activity participants into child (child, daughter, and son), fathers reported participating with their children (unspecified sex of child/ren) in activities in 12.1% (*n* = 105/866) of their activities, with sons in 5.1% of their activities (*n* = 44/866) of the time, and with daughters in 1.2% (*n* = 10/866) of their activities. Table 4 provides frequencies of the activities reported. Similarly, Table 5 provides frequencies of with whom the activities were done.

Cross-tabulation analysis revealed paternal reports of activities with their family (*n* = 344) was most closely related to the paternal reported responsibility of family expenses. The most related variables among the cross-tabulations were family expenses with outside chores (*n* = 27/344; 7.8%; *p* = 0.20) and family expenses with television viewing (*n* = 21/344; 6.1%; *p* = 0.02). The top two associations between paternal reported responsibilities and time spent with any children (*n* = 105) were family expenses and outside chores (*n* = 18/105; 17.1%) and childcare and outside chores (*n* = 9/105; 8.6%). Cross-tabulations between paternal responsibilities and activities with daughters (*n* = 10) revealed sporadic associations between several categories that included cooking (*n* = 1), inside chores (*n* = 1), an outing (*n* = 2), outside chores (*n* = 2), play (*n* = 1), sport (*n* = 1), rest (*n* = 1), and tv (*n* = 1). Cross-tabulations between paternal responsibilities and activities with sons (*n* = 42) revealed a centralization within inside (*n* = 7; 16.7%) and outside chores (*n* = 32; 76.2%).

## 4. Discussion

This study investigated the role of Mexican-heritage fathers, their perceived responsibilities, and their activities. Specifically, this study examined paternal responsibilities and self-reported activities including child co-participation in PA. Previous research has identified mothers as primary caretakers within their homes [26]; however, fathers reported many familial responsibilities as well as activities with their family but few specifically with children.

Fathers reported feeling responsible for the family expenses most often in this sample. This focus on expenses and providing for the family is consistent with research on father priorities in immigrant families [35]. Further, previous literature has found that Latino fathers engaged less in child caregiving than fathers identified as other race and ethnic groups [36]. While financial support is a significant tangible support for child PA [37], many Latino fathers report financial and time constraints related to providing for the family as barriers to being engaged in childcare activities including PA co-participation [25]. This perceived lack of time children may also correspond to the lack of activities reported specifically with their children.

Fathers reported doing most activities alone or with the family as a whole. In past research, fathers who reported being physically active with more family members, as opposed to friends, attained significantly more daily moderate-to-vigorous PA [38]. Latino fathers report wanting to do things with their children and be a role model when it comes to PA, but also report a lack of knowledge or information on how to support healthy eating and activity behaviors for their children [25]. Filling this knowledge or skill gap may provide fathers with the confidence to support healthy child PA behaviors.

After separating children by sex, fathers reported participating in outside chores more often with their sons than with their daughters. However, fathers were more likely to report doing sedentary activities with daughters than with their sons. Our findings reflect previous research showing fathers are more involved in the lives of their sons [39] and that boys are more active than girls in moderate to vigorous activities that include sports [40]. These sex disparities are concerning and could be addressed through co-participation in activities including fathers engaging in outside chores and sport activities with children. Promotion of these activities, with a highlight on co-participation with daughters, would be a simple intervention strategy as fathers within this population often reported doing these activities with their sons.

Positively, fathers within Mexican-heritage culture hold specific cultural beliefs that increase selflessness (familism: placing family interest and development ahead of personal growth [41]), responsibility, and connectedness which could be used alongside educational materials to decrease activities done alone (positive machismo [42,43]). Future programmatic efforts need to be culture specific to incorporate familism and positive machismo concepts within child PA to increase engagement, decrease PA done alone, and help fathers understand their role and importance within child PA as well as ways to engage with their children in PA.

### 4.1. Implications

Building on these results, future intervention development should work with fathers to identify ways of building in more outdoor PA time with children and specifically girls. Paternal presence during outdoor child PA could help decrease feelings of danger that mothers often report as the reason for not allowing outdoor play within their neighborhoods [44], activity in parks, and active transportation [45], specifically for daughters. These are important considerations as researchers have shown that increasing perceptions of safety can increase child PA while also decreasing risk of physical and social disorders [46].

### 4.2. Strengths and Limitations

One strength of this study was the open-ended design of elicitation survey items. This removed researcher bias in an unexplored area by allowing fathers to report what they felt was important without being led by researcher driven motives and/or ideas that are possible when pre-set response items are provided. Intentional and iterative *promotora* feedback during survey development both in content and translation greatly strengthened validity and reliability of our findings. This study also benefited from a robust sample of Mexican-heritage fathers (*n* = 300) which is lacking in current literature. Additionally, incorporating dual coding methods helped to establish validity and reliability. 

Despite the strengths, the survey did not specifically ask fathers to quantify time spent with children. Future work should assess the duration of paternal time with children as it may be underestimated if fathers reported family time and children time together in the current study. Although we cannot disaggregate family time, we were able to compare reported activities between daughters and sons which can be used to inform intervention strategies. 

## 5. Conclusions

This study is an important first step to adding fathers’ perspective to promote their influence to support child PA among Mexican-heritage families. While researchers have primarily focused on mothers’ perceptions and roles, supporting fathers to take an active role in their children’s PA may further promote healthy activity behaviors. Future research should continue to explore paternal beliefs and ideas to better understand the role of fathers in child PA and to create multi-faceted strategies that include both parents. *Promotoras* could serve as interview moderators for the collection of qualitative data that were used for immediate intervention creation. Researchers, with the help of *promotoras*, could then work toward a tailored intervention that met the interests and needs of Mexican-heritage fathers. Providing this cultural tailoring specific to these fathers in this context would be best to promote positive health behaviors.

## Figures and Tables

**Table 1 ijerph-18-08618-t001:** Codes and examples of paternal responsibilities.

Parent Code	Child Code	Examples
Family	Childcare	Help with homework; take them to the doctor; look after his children
	Child physical activity	We play baseball; take children to play sports
	General	Take care of family; supporting them in everything
	Guidance	Education of daughter for her to exercise and eat well; teach them values; advise them in what is good and bad
	Health	Take care of them so they stay healthy; taking care of physical, emotional, and spiritual needs; maintain health of children
	Household tasks	Maintain the yard; help with cleaning the home
	Protection	Protect the children; protected from everything
	Time	Talk with family; be aware of their needs; spend time with them
Monetary	Animals	Maintain animals
	Bills	Pay the utility expenses; pay the bills and rent; buy groceries; pay bills and all accounts
	Family expenses	Provide food; give them what they need, food, clothing; provide what is needed for the family; work to maintain the family
	General	Work and take care that there is nothing missing at home; and for them not to lack anything
	Home	Maintain the home

**Table 2 ijerph-18-08618-t002:** Codes for paternal activities.

Physical Activity	Sedentary
Cooking	Church
Exercise	Leisure
Fish	Read/study
Inside chores	Rest
Outing	Trip
Outside chores	Television
Play	Visiting
Sport	
Work	

**Table 3 ijerph-18-08618-t003:** Paternal responsibilities reported.

Parent Code	Child Code	*n*	%
Family	Child Care	88	14.0
	Child Physical Activity	7	1.1
	General	24	3.8
	Guidance	71	11.3
	Health	23	3.7
	Household Tasks	33	5.3
	Protection	13	2.1
	Time	18	2.9
Monetary	Animals	1	0.2
	Bills	73	11.6
	Family Expenses	164	26.2
	General	76	12.1
	Home	36	5.7

**Table 4 ijerph-18-08618-t004:** Paternal reported activities.

Parent Code	Child Code	*n*	%
Physical Activity	Cooking	30	3.46
	Exercise	6	0.69
	Fish	13	1.50
	Inside chores	141	16.28
	Outing	37	4.27
	Outside chores	359	41.45
	Play	22	2.54
	Sport	21	2.42
	Work	16	1.85
Sedentary	Church	17	1.96
	Leisure	5	0.58
	Read/study	11	1.27
	Rest	39	4.50
	Trip	78	9.01
	Tv	58	6.70
	Visiting	13	1.50

**Table 5 ijerph-18-08618-t005:** Paternal reported of others with whom they did activities.

Whom	*n*	%
General Family	344	39.72
Child (ren)	95	10.97
Daughter	10	1.15
Extended family	11	1.27
Grandkids	3	0.35
Parents	1	0.12
Siblings	4	0.46
Son	42	4.85
Wife	73	8.43
Friend/Other	9	1.04
Alone	274	31.64

## Data Availability

The data analyzed for the current study is not publicly available due to ethical restrictions related to privacy and the consent/assent given by participants at the time of study commencement. An ethically compliant dataset may be made available by the investigators (Sharkey and Umstattd Meyer) on reasonable request and upon approval by the Texas A&M University Institutional Review Board.

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
