# Peer review of "¿Qué Pasa Con Papá? Exploring Paternal Responsibilities and Physical Activity in Mexican-Heritage Families"

_ijerph, 2021, doi:10.3390/ijerph18168618_

Round 1

Reviewer 1 Report

The article is interesting, the topic is actual.

However, more than 50% of the references used are not included in the 5-year interval 2016-2020. I am sure that there is a sufficient amount of modern literature on this issue, the use of which, perhaps, would change the content of this article.

The article is well written, read with interest.

Author Response

A literature review was conducted on this topic with minimal information found especially within this population. All relevant information was included in this article spanning from the 90s to present day. We understand the need for recent literature but this also displays the lack of understanding and gap we are trying to fill through this work and paper.

Reviewer 2 Report

- The title is concise and representative of the whole research. 
- The abstract contains the expected sub-sections: problem, methodology, results and discussion.
- The number of keywords should be increased for better readability and citation of the article. 
- Physical activity (PA) should always be capitalised with an A, as they are acronyms or abbreviations, which become PA. 
- The introduction is interesting, but very short. I recommend delving into more literature, international, recent and prestigious, which has certainly delved into this topic or very similar topics. 
- The conclusions are too short for such a deep problem and such relevant research. They should be expanded, talk about research prospects, point out why this work is original, contributes knowledge and will be replicable by the research community; and relate the conclusions to the theoretical framework cited and commented on. 

Reviewer 3 Report

Exploratory study addresses a gap in the literature. Study methods are described and appropriate. One comment/questions I had was in reference to the use of promotoras (line 78). Sounds like female promotoras were engaged and not males.. while not common, there are males promotors and just curious if this was intentional or based on employed promotores, etc.  Might add 1-2 sentences explaining as some might wonder about the male/female (and particularly important if the selection by sex was intentional). Results and discussion are sufficient. Conclusion and next steps/implications could be strengthened--particularly in light of the weaknesses mentioned--what might be the next steps to explore this topic and maybe even address potential interventions? Also, might think about adding more detail the role of promotoras in the design/implementation of the study.  

Round 2

Reviewer 1 Report

The authors of the article have not changed the list of references, my remark has not been corrected. Still, more than 50% of the links used did not fall within the five-year interval 2016-2020. I am sure that there is enough modern literature on this subject, the use of which, perhaps, would change the content of this article. 

Author Response

While we understand the need to cite recent studies, we have scanned the literature and feel that these are accurate representations of the current body of literature. This being said it further implicates the gap in the literature we are looking to fill as there are not many current articles on this topic. 

Below we provide additional insight as to why the reviewers' described ‘outdated’ references are necessary within this paper. The specific justification for each reference has been provided below.

Citations and justification for all before 2015

5 – one of multiple studies showing that physical inactivity has been linked to childhood disease for years; used to show longitude and scope of research.

8, 10, 11 – three of multiple studies showing the disproportionate disparities faced among rural areas regarding physical activity; used to show longitude and scope of research.

16, 17 – these are very specific to the particular population within this study; little research has been conducted on this particular population. 

18 – one of multiple studies showing scope of research within settings for child PA.

22, 24 – two articles pulled from a published review (reference 20) showing that little attention has been placed on fathers; highlighting purpose and need for this current article.

26, 27 – two of only a few articles speaking about this particular population.

28 – first article published that gives the definition for promotoras.

31, 32, 33 – are the original articles detailing the methodology that was used within this exploratory approach.

35, 36 – two articles found describing priorities of fathers within a similar population to the one currently being studied; no work done within this particular population.

41, 42 – these two articles provided definitions that are paramount to the heritage of this particular population.

Reviewer 2 Report

Congratulations on the improvements made. I think it is a very interesting research that should definitely be published.

Author Response

Thank you for your valuable suggestions and the congratulatory note!

Reviewer 3 Report

Paper is publish ready and contributes to the field. 

Author Response

Thank you for your valuable suggestions and the congratulatory note.